



**Effectiveness evaluation of temporary emission control action in 2016**
**winter in Shijiazhuang, China**
Baoshuang Liu[a], Yuan Cheng[a], Ming Zhou[a], Danni Liang[a], Qili Dai[a], Lu Wang[a], Wei Jin[b], Lingzhi
Zhang[b], Yibin Ren[b], Jingbo Zhou[b], Chunling Dai[b], Jiao Xu[a], Jiao Wang[a], Yinchang Feng[a*], and
Yufen Zhang[a*]
[a] *State Environmental Protection Key Laboratory of Urban Ambient Air Particulate Matter*
*Pollution Prevention and Control, College of Environmental Science and Engineering, Nankai*
*University, Tianjin, 300071, China*
[b] *Environmental Monitoring Station, Shijiazhuang, Hebei, 050023, China*

* Tel./fax: +86 02285358792.
E-mail address: fengyc@nankai.edu.cn (Y. Feng) and zhafox@126.com (Y. Zhang)



**Abstract.**
To evaluate the environmental effectiveness of the control measures for atmospheric pollution
in Shijiazhuang of China, a large-scale controlling experiment for emission sources of atmospheric
pollutants (i.e., a temporary emission control action, TECA) was designed and implemented during
November 1, 2016 to January 9, 2017. Under the unfavorably meteorological conditions, the mean
concentrations of $PM_{2.5}$, $PM_{10}$, $SO_2$, $NO_2$, and chemical species (Si, Al, $Ca^{2+}$, $Mg^{2+}$) in $PM_{2.5}$ during
the control action and heating period (CAHP) still decreased by 15 %, 26 %, 5 %, 19 %, 30.3 %,
4.5 %, 47.0 % and 45.2 %, respectively, compared to the no control action and heating period
(NCAHP); indicating that the control measures of atmospheric pollution in Shijiazhuang were
effective and was in a right direction. Overall, the effects of control measures in suburbs were better
than those in urban area, especially for the control effects of particulate matters sources. The control
effects for emission sources of carbon monoxide (CO) were not apparent during the TECA period,
especially in suburbs, which is likely due to the increasing usage of domestic coal in suburbs along
with the temperature decreasing.
The results of PMF analysis showed that crustal dust, secondary sources, vehicle emissions, coal
combustion and industrial emissions were major $PM_{2.5}$ sources. Compared to the whole year, the
contributions of coal combustion to $PM_{2.5}$ increased significantly during the CAHP and after control
action (ACA); while the contribution proportions of crustal dust and vehicle emissions to $PM_{2.5}$
decreased apparently during the CAHP. The contribution concentrations and proportions of crustal
dust and vehicle emissions to $PM_{2.5}$ during the CAHP also decreased significantly compared to ACA.
The pollutant's emission sources during the CAHP were in effective control, especially for crustal
dust and vehicles. While the necessary coal heating for cold winter and the unfavorably
meteorological conditions had an offset effect on the control measures for emission sources to some
extent. Meanwhile, our results also illustrated that the discharge of pollutants was still enormous
even under such strict control measures.
The backward trajectory and potential source contribution function (PSCF) analysis in the light
of atmospheric pollutants suggested that the potential sources-areas mainly concentrated in
surrounding regions of Shijiazhuang, i.e., south of Hebei, north of Henan and Shanxi. The regional
nature of the atmospheric pollution in Northern China Plain revealed that there is an urgent need for
making cross-boundary control policy except for local control-measures given the high background



level of pollutants.

The TECA is an important practical exercise but it can't be advocated as the normalized control

measures for atmospheric pollution in China. The direct cause of atmospheric pollution in China is
the emission of pollutants exceeds the air environment's self-purification capacity, and the essential
reason is unreasonable and unhealthy pattern for economic development of China.

**Keywords:** Atmospheric pollutants; Effectiveness evaluation; Control action; PMF; PSCF



## 1    Introduction


As a consequence of rapid industrialization and urbanization, China has been suffering from

air quality degradation in recent years (Fu et al., 2014; Gao et al., 2015; Han et al., 2014; Hao et al.,
2017; Zhao et al., 2011). Frequently occurred severe haze is featured by long duration, extensive
coverage and sharply-increasing particulate concentration (Jiang and Xia, 2017; Tao et al., 2014;
Wang et al., 2016a; Zhang et al., 2015a). It has been suggested that severe haze pollution increase
the risk of respiratory and cardiovascular diseases (Chen et al., 2013; Gao et al., 2015; Pan et al.,
2014; Zhang et al., 2014a; Zhou et al., 2015). On the basis of previous statistics, there are four haze-
prone city clusters in China, including Beijing-Tianjin-Hebei region, Yangtze River Delta, Pearl
River Delta and Sichuan Basin (Bi et al., 2014; Chen et al., 2016a; Fu et al., 2014; Fu and Chen,
2017; Li et al., 2016b; Tao et al, 2013a; Wang et al., 2015b; Wu et al., 2008; Zhang et al., 2015b).
In recent years, the role of particulates in hazy events has been becoming more and more prominent.
The particulates can be discharged from varieties of sources or formed by physicochemical/aqueous-
oxidation reactions between gaseous precursors, which have significant negative effects on climate,
atmospheric visibility and public health (Chen et al., 2015; Fu and Chen, 2017; Lee et al., 2015;
Quinn and Bates, 2003; Shen et al., 2015; Tai et al., 2010; Zhang et al., 2010). The high observed
concentrations of fine particles and prolonged haze events have occurred frequently during autumn
and winter, and covered large regions in China. In some cases, the instantaneous mass concentration
of $PM_{2.5}$ had reached 1000 $\mu g/m^3$ (Qin et al., 2016; Zhang et al., 2014b), which caused the extensive
concern from citizens and government agencies.

Confronted with severe air pollution and degradation of air quality, the government has carried

out a variety of control measures in recent years, including odd-and-even license plate rule
(http://www.sjz.gov.cn/col/1274081553614/2016/11/17/1479391129628.html),                    mandatory
installation of desulfurization, denitration and other pollution-controlling facilities in factories (Liu
et al., 2017a; Ma et al., 2015; Peng et al., 2017) and on-line monitoring system structure plan in
construction sites, etc. The atmospheric quality in China has been notably improved so far. From
2013 to 2016, the concentrations of atmospheric pollutants showed a decreased trend, and the annual
mean concentrations of $PM_{2.5}$, $PM_{10}$, $SO_2$ and $NO_2$, in 2016 reached up to 50 $\mu g/m^3$, 85 $\mu g/m^3$, 21
$\mu g/m^3$  and  39  $\mu g/m^3$,  respectively,  and  significantly  lower  than  those  in  2013
(http://www.zhb.gov.cn/hjzl/zghjzkgb/lnzghjzkgb/).  However,  the  concentrations  of  $PM_{2.5}$  and



PM$_{10}$ in 2016 were still 1.4 and 1.2 times higher than the national ambient air quality standard
(NAAQS) (GB3095-2012 Grade II, PM$_{2.5}$: 35 μg/m$^3$, PM$_{10}$:70 μg/m$^3$). Note that the concentrations
of PM$_{2.5}$ and PM$_{10}$ during Beijing-Tianjin-Hebei region were up to 71 μg/m$^3$ and 119 μg/m$^3$ in 2016,
and 2.0 and 1.7 times higher than the NAAQS, respectively. Therefore, China still has a lot of work
to do to improve the national air quality.

Over the last decade, Chinese government has implemented stricter control-measures for

emission sources during multiple international events held in China than normal times (Chen et al.,
2016b; Guo et al., 2013; Liu et al., 2013; Sun et al., 2016; Wang et al., 2010; Wang et al., 2017). For
instance, the first attempt took place during the Beijing 2008 Olympic Games (Guo et al., 2013).
Drastic control actions were executed to cut down the emissions of atmospheric pollutants from
motor vehicles, industries and building construction activity (UNEP, 2009; Wang et al, 2009a; Wang
et al., 2010). UNEP (2009) suggested that the concentration of PM$_{10}$ in Beijing was reduced by 20 %
due to the emission reduction measures. Liu et al. (2013) reported that the concentrations of SO$_2$,
NO$_2$, PM$_{10}$ and PM$_{2.5}$ were reduced by 66.8 %, 51.3 %, 21.5 % and 17.1 %, respectively, during the
2010 Asian Games in Guangzhou of China, and during which stricter control measures for emission
sources were implemented. Furthermore, further stricter controls for emission sources were
implemented in both Beijing and its surrounding regions during the 2014 Asia-Pacifc Economic
Cooperation (APEC) summit and Parade. Compared to no-control during APEC and Parade, a
decreasing trend with 51.6~65.1 % and 34.2~64.7 % of PM$_{2.5}$ concentrations during the control
period was reported (Wang et al., 2017). Eventually, all the efforts led to a blue-sky days during the
APEC, which was acknowledged as "APEC Blue" (Wang et al., 2016b). As we can see that the air
quality can be improved in response to stricter emission controls in international events held in
China. However, once these stricter control-measures of emission sources were repealed, and the
air        quality        would        be        deteriorated        subsequently
(http://www.mep.gov.cn/gkml/hbb/qt/201412/t20141218_293152.htm),        indicating        that        the
prevention and control of air pollution in China still had a long way to go.

Shijiazhuang (38.03º N, 114.26º E), a hinterland city of Northern China Plain with a high

population density, is an important city in Beijing-Tianjin-Hebei region (Sun et al., 2013). The rapid
industry development has a great contribution to this city's economic growth and degradation of air
quality at the same time (Du et al., 2010; Li et al., 2015; Yang et al., 2015, 2016a). Shijiazhuang has



been one of the cities with the most serious air pollution in the world
(https://www.statista.com/chart/4887/the-20-worst-cities-worldwide-for-air-pollution/), and
deteriorating air quality poses a great risk to public health (http://www.who.int/ceh/risks/cehair/en/),
as well as drags on the expansion of the economy. The government of Shijiazhuang has adopted a
variety of control measures (http://www.sjzhb.gov.cn/), however, it seems that the improvement in
air quality of Shijiazhuang is not go into effect so far, and the atmospheric pollution is still heavy.
In 2016, the annual concentrations of $PM_{2.5}$ and $PM_{10}$ in Shijiazhuang reached up to 70 μg/m$^3$ and
123 μg/m$^3$, respectively, which were 2.0 and 1.8 times higher than the NAAQS (GB3095-2012
Grade II) (http://www.zhb.gov.cn/hjzl/tj/201706/_t20170606_415527.shtml). Especially in the
heating period in winter, the degree of atmospheric pollution in Shijiazhuang was even more serious.
The effectiveness of control measures has been queried in recent years. Therefore, based on previous
examples of APEC, Parade and the Asian Games, etc., a large-scale controlling experiment for
atmospheric pollutants sources (i.e., TECA) was designed and implemented to investigate whether
control measures in Shijiazhuang are effective for the atmospheric pollution. The experiment was
carried out in Shijiazhuang during November 1 2016 to January 9 2017, during which more stringent
control measures of atmospheric pollution than usual were put into practice. Then, by combining of
the changes of atmospheric pollutants concentrations, emission source contributions and other
factors such as meteorological conditions, regional transmission, etc., the effectiveness of control
measures was evaluated before and after the control measures were taken.
**2 Materials and Methods**
**2.1 Site description**
Shijiazhuang city is located in the east of Taihang Mountain in north of China (Fig. 1), and the
urban area is 15848 km$^2$, with a population of more than 10 million in 2016. Shijiazhuang is a large
industrial city that is famous for raw materials, energy production and steel, power, and cement
industries. The number of vehicles is more than 2.0 million until 2016. Shijiazhuang has a typical
temperate and monsoonal climate with four clearly distinct seasons, with northeasterly,
southeasterly and northwesterly winds prevailed during the TECA period (Fig. S1). The mean wind
speed was 0.6 m/s, and the average temperature was 14.9 ℃ during the TECA period. The mean
relative humidity was up to 76.5 %, and the mean height of mixed layer was 509 m during the TECA
period. The meteorological conditions during the four stages of the TECA period in Shijiazhuang



were shown in Table 1.

The seven monitoring sites including Twenty-second Middle School (TSMS), High-tech Zone

(HTZ), Great Hall of the People (GHP), Century Park (CP), Water Source Area in the Northwest
(WSAN), University Area in the Southwest (UAS) and Staff Hospital (SH) are located in urban area
of Shijiazhuang. While other seventeen sites including Fenglong Mountain (FLM), Gaoyi (GY),
Gaocheng (GC), Xingtang (XT), Jinzhou (JZ), Jingxing Mining District (JXMD), Lingshou (LS),
Luquan (LQ), Luancheng (LC), Pingshan (PS), Shenze (SZ), Wuji (WJ), Xinle (XL), Yuanshi (YS),
Zanhuang (ZH), Zhaoxian (ZX) and Zhengding (ZD) are suited in suburbs of Shijiazhuang. The
more details were shown in Table S1.

----

**Fig. 1.** Maps of the online monitoring stations and the filter membrane sampling sites in Shijiazhuang. The 24 online
monitoring stations mainly include Twenty-second Middle School (TSMS), Fenglong Mountain (FLM), High-tech
Zone (HTZ), Great Hall of the People (GHP), Century Park (CP), Water Source Area in the Northwest (WSAN),
University Area in the Southwest (UAS), Staff Hospital (SH), Gaoyi (GY), Gaocheng (GC), Xingtang (XT), Jinzhou
(JZ), Jingxing Mining District (JXMD), Lingshou (LS), Luquan (LQ), Luancheng (LC), Pingshan (PS), Shenze (SZ),
Wuji (WJ), Xinle (XL), Yuanshi (YS), Zanhuang (ZH), Zhaoxian (ZX) and Zhengding (ZD). The filter membrane
sampling sites are mainly located in TSMS, LQ and LC.
**Table 1.** The meteorological conditions during the four stages (NCANHP, NCAHP, CAHP and ACA) of the TECA
period in Shijiazhuang.

----

**2.2 Sampling and Analysis**
**2.2.1 Sampling**

From November 1, 2016 to January 9, 2017, the concentrations of $PM_{2.5}$, $PM_{10}$, $SO_2$, $NO_2$, CO,

$O_3$ and synchronous meteorological conditions (temperature, relative humidity, wind speed and
wind direction) were monitored in the 24 monitoring sites belonged to national, provincial and city
controlling points (Fig. 1). The more details about monitoring instruments were described in Table
S2. The heights of mixed layer were measured with a lidar scanner (AGHJ-I-LIDAR (HPL)), which
was set at an atmospheric gradient monitoring station in Shijiazhuang near CP site (Fig. 1), and
more details were shown in supplemental material. The $PM_{2.5}$ filter membrane samples were
collected in TSMS, LQ, and LC sites from November 24, 2015 to January 9, 2017. Three sampling
sites were set on the rooftops of buildings at 12-15 meters above ground level. Meanwhile, the
parallel samples and the field blanks were also collected at each site. More details about filter
membrane sampling were shown in Table S3. Before sampling, the quartz filter membranes (47 mm



in diameter, Whatman, England) and polypropylene filter membranes (47 mm in diameter, Beijing
Synthetic Fiber Research Institute, China) were baked in the oven at 500 °C and 60 °C, respectively.
All the filter membranes after sampling were stored at 4 ℃ before subsequent gravimetric and
chemical analysis to improve the accuracy of experimental results.
**2.2.2 Gravimetric and Chemical analysis**
A 24-hour equilibrium process of $PM_{2.5}$ filter membranes was performed at a condition of
constant temperature ($20\pm1$ ℃) and humidity (45-55 %) before gravimetric analysis. For the
gravimetric analysis, all the filter membranes were weighted twice on a microbalance with
resolution of 0.01 mg (Mettler Toledo, XS105DU) before and after sampling. An electrostatic
eliminating device was applied to ensure the accuracy of gravimetric results.
After the gravimetric analysis, the quartz filter membranes which carried atmospheric
particulates were used to analyze water-soluble ions by Ion chromatography (Thermo Fisher
Scientific, Dionex, ICS-5000+). One-eighth of the filter membrane was cut up and put into a 25 mL
glass tube with 20 mL ultrapure water. After 1-hour ultrasonic extraction and 3 minutes
centrifugalization, the supernatant was filtered with disposable filter head (0.22 μm) for subsequent
instrumental analysis. The ions analyzed included $SO_4^{2-}$, $NO_3^-$, $Cl^-$, $NH_4^+$, $K^+$, $Ca^{2+}$, $Na^+$ and $Mg^{2+}$,
and more details were shown in Figs. S2 and S3. Prior to the ions detection, standard solutions were
prepared and detected for over three times and low relative standard deviations (RSD) were obtained.
Analytical quantification was carried out by using calibration curves made from standard solutions
prepared.
Polypropylene filter membranes were used for elemental analysis by inductively coupled
plasma–mass spectrometry (ICP-MS, Agilent 7700x). Perchloric acid-nitric acid digestion method
was applied for the pretreatment of filter membranes. Aggregately, 10 elemental species (Al, Si, Ti,
Cr, Mn, Fe, Cu, Zn, As and Pb) were determined. The detection limits of all the elements were
shown in Table S4. For quality assurance and quality control (QA/QC), standard reference materials
were pre-treated and analyzed with the same procedure, with the recovered values for all the target
elements falling into the range or within 5 % of certified values.
The OC and EC were determined on a 0.558 $cm^2$ quartz filter membrane punch by Desert
Research Institute (DRI) Model 2001 Thermal/Optical Carbon Analyzer with IMPROVE A
thermal/optical reflectance (TOR) protocol. The quartz filter membrane was heated stepwise to



temperatures of 140 °C, 280 °C, 480 °C and 580 °C in a non-oxidizing helium (He) oven to analyse
OC1, OC2, OC3 and OC4, respectively. Then, the oven was added to an oxidizing atmosphere of
2 % oxygen ($O_2$) and 98 % He, and the quartz filter membrane was gradually heated to 580 °C,
780 °C and 840 °C to analyse EC1, EC2 and EC3, respectively. The POC is defined as the carbon
combusted after the initial introduction of oxygen and before the laser reflectance signal achieves
its original value and the POC is specified as the fraction of OC. According to the IMPROVE A
protocol,    OC    is    defined    as    OC1+OC2+OC3+OC4+POC,    and    EC    is    defined    as
EC1+EC2+EC3−POC. For QA/QC, we carried out the measurement with the field blank filter
membranes, standard sucrose solution and repeated analysis in the study. During each season, the
field blanks were sampled and the particulate samples have been corrected by the average
concentration of the blanks. For checking the precision of instrument, a replicate sample was
analysed for every 10 samples, and the standard deviation $<\pm 5$ % was accepted. The method
detection limits (MDLs) of OC and EC are 0.45 and 0.06 $\mu g/cm^2$, respectively.

**226    2.3 PMF model**

PMF model can decompose a matrix of sample data (X) into two matrices: source profile (F)

and source contribution (G), in terms of observations at the sampling sites (Paatero and Tapper,
1994). The principle of PMF model can be described by:
$$X_{ij} = \sum_{k=1}^{p} g_{ik} f_{kj} + e_{ij} \qquad \textbf{(1)}$$
where $X_{ij}$ represents concentration of the $j^{th}$ species in the $i^{th}$ sample, $g_{ik}$ represents the contribution
of the $k^{th}$ source to the $i^{th}$ sample, $f_{kj}$ represents the source profile of $j^{th}$ species from the $k^{th}$ source,
$e_{ij}$ represents the residual for the $j^{th}$ species in the $i^{th}$ sample, and $p$ represents the number of sources.

PMF can identify emission sources of $PM_{2.5}$ without source profiles. Data below MDLs are

retained for using in PMF model with the related uncertainty adjusted in terms of the characteristics
that PMF model admits data to be signally weighed. To assess the stability of the solution, the object
function Q can be allowed to review the distribution of each species, which is expressed by:
$$Q = \sum_{i=1}^{n} \sum_{j=1}^{m} \left[ \frac{x_{ij} - \sum_{k=1}^{p} g_{ik} f_{kj}}{\mu_{ij}} \right]^2 \qquad \textbf{(2)}$$
where $\mu_{ij}$ represents the uncertainty of $j^{th}$ species in the $i^{th}$ sample, which is applied to weight the
observations that include the sampling errors, missing data, detection limits and outliers.

The purpose of PMF model was to minimize the function (Eq. (2). Data below MDLs were





retained and their uncertainties were set to 5/6 of the MDLs. Missing values were replaced by the
median concentration of a given species, with an uncertainty of four times the median (Brown et al.,
2015). Values that were larger than the MDLs, the calculation of uncertainty was in terms of a user
supplied fraction of the concentration and MDLs, and the error fraction was suggested as 10 % by
Paatero (2000). Uncertainty was described by:
$$\text{Uncertainty} = \sqrt{(Error\ Fraction \times concentration)^2 + (0.5 \times MDL)^2} \qquad (3)$$
In this study, EPA PMF 5.0 model was used to identify the $PM_{2.5}$ sources in Shijiazhuang city.
Based on the field investigation and change of $Q$ values, and finally, five factors were chosen in
PMF analysis. When five factors were chosen and input in PMF model, and the calculated $Q$ value
(5162) from PMF model was close to theoretical values (5045). The observed $PM_{2.5}$ concentrations
and calculated $PM_{2.5}$ concentrations from PMF model showed high correlations ($r = 0.96$) (Fig. S4).
S/N is the signal-to-noise ratio, which is used to address weak and bad variables when running PMF
model (Paatero and Hopke, 2003). The signal vector is identified as S and the noise vector is
identified as N. Next, S/N is defined as Eq. (4). Variables with S/N $\leq 0.2$ were removed from the
analysis, while weak variables ($0.2 \leq$ S/N $\leq 2.0$) were down-weighted (Ancelet et al., 2012). S/N of
As, Ti and Cr were lower than 1.0 in this study, and these species were set as weak variables.
$$S/N = \sqrt{\sum {s_i}^2 / \sum {n_i}^2} \qquad (4)$$
where $i$ represents the chemical species in $PM_{2.5}$.
**2.4 Backward trajectory and PSCF model**
In this study, the 72-h backward trajectory arriving in Shijiazhuang (38.05˚ N, 55.2˚ E) was
calculated at 1-h intervals during the CAHP by the Hybrid Single Particle Lagrangian Integrated
Trajectory (HYSPLIT) model. The final global analysis data were produced from the National
Center for Environmental Prediction's Global Data Assimilation System wind field reanalysis
(http://www.arl.noaa.gov/). The model was run 4 times per day at starting times, i.e., 0:00, 06:00,
12:00, 18:00 LT; the starting height was set at 100 m above the ground. The PSCF model was used
to identify the potential sources-areas in terms of the HYSPLIT analysis. The study region was
divided into $i \times j$ small equal grid cells. The trajectory clustering and PSCF model were performed
by using the GIS-based software TrajStat (Liu et al., 2017a; Wang et al., 2009b). The PSCF value
was defined as:



$$PSCF = \frac{m_{ij}}{n_{ij}} \qquad (5)$$

where $i$ and $j$ were the latitude and longitude indices, $n_{ij}$ represented the number of endpoints that
fell in the $ij$ cell, and $m_{ij}$ was the number of endpoints in the same cell that were related to the
samples that were greater than the threshold criterion.

Based on the NAAQS (GB3095-2012 guideline value (24 h) of Grade II), the criterion values

of $PM_{2.5}$, $PM_{10}$, $NO_2$, CO were set to 75 μg/m$^3$, 150 μg/m$^3$, 80 μg/m$^3$ and 4 mg/m$^3$, respectively. The
criterion values of $SO_2$ and $O_3$ were set to 68 μg/m$^3$ and 15μg/m$^3$ respectively, in terms of the
average during the CAHP. When $n_{ij}$ is smaller than three times the grid average number of trajectory
endpoint ($n_{ave}$), a weighting function $W(n_{ij})$ was used to reduce uncertainty in cells (Dimitriou et al.,
2015). The weighting function was defined by:
$$WPSCF_{ij} = \frac{m_{ij}}{n_{ij}} * W(n_{ij}) \qquad (6)$$

$$W(n_{ij}) = \begin{cases} 1.00, 3n_{ave} < n_{ij} \\ 0.70, 1.5n_{ave} < n_{ij} \leqslant 3n_{ave} \\ 0.40, n_{ave} < n_{ij} \leqslant 1.5n_{ave} \\ 0.20, n_{ij} \leqslant n_{ave} \end{cases}$$

(7)

The studying field ranged from 33˚ N to 51˚ N, and 97˚ E to 121˚ E, and the region that was

covered by the backward trajectories was divided into 432 grid cells of 1.0˚ × 1.0˚. The total number
of endpoints during the CAHP was 12672. Accordingly, there was an average of 5 trajectory
endpoints in per cell ($n_{ave}$ = 5).
**2.5 Measures taken in the controlling experiment**

The measures taken in the controlling experiment began on November 18, 2016 and ended on

December              31,              2016              in              Shijiazhuang
(http://www.sjz.gov.cn/col/1274081553614/2016/11/17/1479391129628.html). The measures taken
in the control action were mainly aimed at controlling emission sources of atmospheric pollutants
in Shijiazhuang, which mainly included five aspects: (1) reduce the usage of coal, (2) decrease
industrial production, (3) inhibition of dust emission, (4) driving restriction, and (5) prohibit open
burning. The more details were described in supplemental material.

Actually, a total of 1543 enterprises were shut down in the whole city of Shijiazhuang during

the control action period, including pharmaceutical, steel, cement, coking, casting, glass, ceramics,
calcium and magnesium, sheet, sand and stone processing, stone processing and other industries.



The situation of specific closed-enterprises in different districts and counties is shown in Table S5.
In closed enterprises in Shijiazhuang, the number of mining and stone processing enterprises was
the largest, which was up to 733 and account for 48 % of all the closed enterprises. The numbers of
casting and building materials enterprises were up to 297 and 227, respectively, accounting for 19 %
and 15 % of the all, respectively. In addition, 64 enterprises related to pharmaceutical industry were
halted only for the VOC technology, and the 17 enterprises related to chemical industry must stop
production. The numbers of closed enterprises for cement and calcium/magnesium industry were
up to 49 and 40, respectively. The number of closed factories related to furniture and tanneries was
43, and the numbers of closed steel and coking enterprises were up to 4 and 7, respectively.
The average value of daily social-electricity consumption from November 18 to December 31,
2016 was 103,470,000 kW • h (Fig. S5), which declined 10 % compared to that of daily social-
electricity consumption from November 1 to 17, 2016, and declined 6 % compared to that of daily
social-electricity consumption during the same period in 2015. Restriction of motor vehicles based
on odd-and-even license plate rule in urban area of Shijiazhuang resulted in the decrease of the
average traffic-flow on arterial roads, which reduced about 30 % compared to before the control
action (Fig. S6). The dust emission can be reduced about 390 tons per day by a series of dust control-
measures. Compared to before the control action, the daily emissions of $SO_2$, $NO_x$, smoke dust and
VOCs reduced about 20 %, 33 %, 15 % and 7 %, respectively, during the control action period, on
the basis of the statistics on pollutants emission inventories.
**3 Results and discussion**
**3.1 Variations of atmospheric pollutants concentrations**
**3.1.1 Temporal trend**
The time series of atmospheric pollutants concentrations during the TECA period are shown in
Fig. 2. The average concentrations of $PM_{2.5}$ and $PM_{10}$ during the TECA period in Shijiazhuang were
up to 181 μg/m³ and 295 μg/m³, respectively, which were 5.2 and 3.2 times than the Grade II limit
values in the NAAQS. The ratio of $PM_{2.5}/PM_{10}$ reached up to 0.62 during the TECA period,
indicating that the fine particulate dominated on the particulate pollution in Shijiazhuang. The mean
concentration of $PM_{2.5}$ during the TECA period was significantly higher than those of winter in
Beijing (95.50 μg/m³), Tianjin (144.6 μg/m³), Hangzhou (127.9-144.9 μg/m³), Heze (123.6 μg/m³)
and Xinxiang (111 μg/m³) (Cheng et al., 2015; Gu et al., 2011; Liu et al., 2015; Liu et al., 2017a;




328 Feng et al., 2016), and lower than those of winter in Handan (240.6 μg/m$^3$) and Xian (266.8 μg/m$^3$)

329 (Meng et al., 2016; Zhang et al., 2011). Additionally, the NAAQS (GB3095-2012 Grade II) values

330 of $SO_2$, $NO_2$, CO and $O_3$ were 60 μg/m$^3$, 40 μg/m$^3$, 4 mg/m$^3$ and 160 μg/m$^3$, respectively. During

331 the TECA period, the average concentration of $SO_2$ (60 μg/m$^3$) could meet the NAAQS, and that of

332 $NO_2$ (81 μg/m$^3$) was far exceed the NAAQS; while those of CO (3.4 mg/m$^3$) and $O_3$ (15 μg/m$^3$)

333 were less than the NAAQS.

334  As well known, the date of coal-fired heating in Shijiazhuang began in November 15, 2016

335 (http://www.sjz.gov.cn/col/1451896947837/2016/10/28/1477635691926.html). Depending on the

336 changes of atmospheric pollution sources, the timeline of the TECA was divided into four stages:

337 stage 1: no control action and no heating period (NCANHP), ranging from November 1 to 14, 2016;

338 stage 2: no control action and heating period (NCAHP), ranging from November 15 to 17, 2016;

339 stage 3: control action and heating period (CAHP), ranging from November 18 to December 31,

340 2016; stage 4: after control action (ACA), ranging from January 1 to 9, 2017.

341  During the TECA period, the variations of atmospheric pollutants concentrations were mainly

342 affected by the heating for cold winter and the control measures of the control action except for the

343 meteorological conditions. Therefore, we defined the following equations to evaluate the effects of

344 the heating and control action, respectively, based on the atmospheric pollutants concentrations

345 during the different stages of TECA (i.e., NCANHP, NCAHP, CAHP and ACA).

346   $$P_{i-heating} = \frac{(C_{i-NCAHP} - C_{i-NCANHP}) \times 100}{C_{i-NCANHP}} \quad \textbf{(8)}$$

347   $$P_{i-action} = \frac{(C_{i-NCAHP} - C_{i-CAHP}) \times 100}{C_{i-NCAHP}} \quad \textbf{(9)}$$

348 where $P_{i-heating}$ represents the increasing percentage (%) of atmospheric pollutant concentration

349 because of the effects of heating for cold winter; $P_{i-action}$ represents the decreasing percentage (%)

350 of atmospheric pollutant concentration because of the control action; $C_{i-NCANHP}$ represents the

351 concentration (μg/m$^3$, CO: mg/m$^3$) of atmospheric pollutant during the no-control action and no-

352 heating period; $C_{i-NCAHP}$ represents the concentration (μg/m$^3$, CO: mg/m$^3$) of atmospheric pollutant

353 during the no-control action and heating period; $C_{i-CAHP}$ represents the concentration (μg/m$^3$, CO:

354 mg/m$^3$) of atmospheric pollutant during the control action and heating period.

355  During the NCANHP, the mean concentrations of $PM_{2.5}$ and $PM_{10}$ were 156 μg/m$^3$ and 253

356 μg/m$^3$ in Shijiazhuang, respectively. With the beginning of heating, the mean concentrations of



PM$_{2.5}$ and PM$_{10}$ increased 44 μg/m$^3$ and 64 μg/m$^3$ during the NCAHP, respectively, and the P$_{PM2.5-}$
$_{heating}$ and P$_{PM10-heating}$ values were up to 28 % and 25 % (Fig. 3 and Fig. 4). However, during the
CAHP, the mean concentrations of PM$_{2.5}$ and PM$_{10}$ were 185 μg/m$^3$ and 291 μg/m$^3$, respectively,
which decreased by 15 % and 26 % compared to the NCAHP. And the P$_{PM2.5-action}$ and P$_{PM10-action}$
values are 8 % and 8 %, respectively. During the ACA, the concentrations of PM$_{2.5}$ and PM$_{10}$ are
227 μg/m$^3$and 383 μg/m$^3$, respectively, which increased significantly by 42 μg/m$^3$ and 92 μg/m$^3$
compared to the CAHP. The variations of SO$_2$ and NO$_2$ concentrations during different stages of
TECA were similar to those of PM$_{2.5}$ and PM$_{10}$ concentrations. The P$_{SO2-heating}$ and P$_{NO2-heating}$ values
were 50 % and 33 %, respectively, and the P$_{SO2-action}$ and P$_{NO2-action}$ values were 5 % and 19 %. Well
known that the NO$_2$ is mainly derived from the vehicle exhaust (Liu et al., 2017b). Therefore, the
control effect of motor vehicles was remarkable during the CAHP in Shijiazhuang. Note that the
mean concentration of CO in Shijiazhuang city varied from 2.2 mg/m$^3$ during the NCANHP to 5.5
mg/m$^3$ during the ACA period, which showed an increasing tendency (Fig. 3). Because CO was
mainly produced from the uncompleted combustion of fossil fuels, so the usage of domestic coal
might be increasing with the gradual decrease of temperature from the NCANHP (8.4 ℃) to the
ACA period (0.7 ℃) (Table 1). Meanwhile, it can also be inferred that the control of domestic coal
during the TECA period in Shijiazhuang city performed little efficiency. Because of the lack of
emission inventories for domestic coal or small-boiler coal in Shijiazhuang, so that the control
measures were less targeted. Additionally, the concentrations of O$_3$ during different stages of TECA
were lower compared to other pollutants (Figs. 2 and 3). Overall, the control measures of emission
sources in Shijiazhuang during the TECA period were go into effect, while the coal heating for cold
winter and the unfavorable meteorological-conditions during the CAHP had an offset effect on the
efforts of control measures for pollutant sources to some extent. The average wind speed during the
CAHP (0.4 m/s on average) was lower than those during the other stages of the TECA period (0.5-
0.7 m/s on average) (Table 1), and the wind directions were changeable (Fig. S1), which was in
favor of the accumulation of atmospheric pollutants, and thus causing the concentrations of
atmospheric pollutants to increase during the CAHP. Note that the heights of mixed layer showed
an apparently decreasing tendency from the NCANHP (540 m on average) and the NCAHP (590 m
on average) to the ACA (431 m on average), and the height of mixed layer during the CAHP was
only 474 m on average (Table 1). The decrease in the height of mixed layer can cause the





concentrations of atmospheric pollutants near the ground to be compressed significantly and
enhanced subsequently. In addition, during the CAHP, the multidirectional air-masses that were
mainly originated from the Beijing-Tianjin-Hebei and its surrounding areas (e.g. Henan,
Shandong and south of Hebei) displayed an overlap with each other in Shijiazhuang (Fig. S7),
and further aggravate the level of air pollution in Shijiazhuang. Furthermore, the effects of
control measures for domestic coal might be worse during the CAHP.

----

**Fig. 2.** The variations of atmospheric pollutants concentrations during the four stages (NCANHP, NCAHP, CAHP
and ACA) of the TECA period in Shijiazhuang.
**Fig. 3.** The concentrations variations of $PM_{2.5}$, $PM_{10}$ and gaseous pollutants during the four stages (NCANHP,
NCAHP, CAHP and ACA) of the TECA period in Shijiazhuang.
**Fig. 4.** The $P_{i\text{-heating}}$ and $P_{i\text{-action}}$ of $PM_{2.5}$, $PM_{10}$ and gaseous pollutants ($SO_2$, $NO_2$, CO and $O_3$) calculated by equation
(8) and (9) in urban area and suburb in Shijiazhuang.

----

**3.1.2 Spatial variation**
The concentrations variations of $PM_{2.5}$, $PM_{10}$ and related gaseous pollutants ($SO_2$, $NO_2$, CO
and $O_3$) during four stages (NCANHP, NCAHP, CAHP and ACA) in urban area and suburb in
Shijiazhuang are shown in Figs. 3 and 5. During the NCANHP, the average concentrations of $PM_{2.5}$
in urban area and suburb were 166 $\mu g/m^3$ and 152 $\mu g/m^3$, respectively. The concentrations of $PM_{2.5}$
in urban area and suburb increased significantly during the NCAHP (t-test, $p<0.01$). The meanly
increased concentration of $PM_{2.5}$ (46 $\mu g/m^3$) in urban area was higher than that of in suburb (43
$\mu g/m^3$), but the value of $P_{PM2.5\text{-heating}}$ in suburb (29 %) was higher than that in urban area (27 %) (Fig.
4). Note that the mean concentration of $PM_{2.5}$ in urban area was up to 243 $\mu g/m^3$ during the CAHP,
which showed an increasing tendency, and the $P_{PM2.5\text{-action}}$ value was -15 % (Fig. 4), likely due to
the unfavorably meteorological conditions such as lower wind speed (0.4 m/s) and lower height of
mixed layer (474 m), etc. (Table 1 and Fig. S7). Conversely, compared to the NCAHP, the
concentrations of $PM_{2.5}$ in suburb (a mean of 161 $\mu g/m^3$) decreased significantly during the CAHP
(t-test, $p<0.01$), and the $P_{PM2.5\text{-action}}$ was up to 18 % (Fig. 4), indicating the control measures of $PM_{2.5}$
sources in suburb might be more effective than urban area. The tendency of $SO_2$ concentrations
during different stages of TECA (except the ACA period) was similar to that of $PM_{2.5}$. The $P_{SO2-}$
$_{heating}$ and $P_{SO2\text{-action}}$ values in urban area were up to 58 % and -4 %, respectively, and were up to 47 %
and 8 % in suburb during the TECA period (Fig. 4). However, the concentrations of $SO_2$ in urban



area and suburb decreased remarkably during the ACA compared to the CAHP (t-test, $p<0.01$),
probably due to the effective control measures.
During the NCANHP, the average concentrations of $PM_{10}$ in urban area and suburb were 280
and 242 μg/m$^3$, respectively. Then, the meanly increased concentrations in urban area and suburb
were up to 65 and 64 μg/m$^3$ during the NCAHP, which were comparable with each other.
Nevertheless, the mean $P_{PM10-heating}$ value in suburb was higher (26 %) than that in urban area (23 %)
(Fig. 4). During the CAHP, the meanly decreased concentration of $PM_{10}$ in urban area was 1 μg/m$^3$,
and apparently lower than that of suburb (36 μg/m$^3$), as well as the mean $P_{PM10-action}$ values in urban
area and suburb were 0.4 % and 12 %, respectively (Fig. 4). It can be seen that the control of $PM_{10}$
sources in suburb was more effective compared to urban area, in case of exclusion of unfavorably
meteorological conditions (Table 1 and Fig. S7), probably related to more than 700 enterprises
closed down which mainly carried out ore mining and stone processing in suburb (Tables S1 and
S5). The tendency of $NO_2$ concentrations in urban area and suburb was similar to that of $PM_{10}$ during
different stages of TECA period. The mean $P_{NO2-heating}$ values in urban area and suburb were up to
31 % and 34 %, respectively; while the mean $P_{NO2-action}$ values in urban area and suburb were up to
17 % and 21 %, respectively. Note that the concentrations of CO in urban area and suburb showed
an increasing tendency from the NCANHP (2.1-2.4 mg/m$^3$) to the ACA period (5.5 mg/m$^3$) (Fig. 3).
The $P_{CO-heating}$ and $P_{CO-action}$ values in urban area were 22 % and -15 %, respectively, while those in
suburb were 32 % and -20 % during the TECA period. In addition, as shown in Fig. 5, the
concentrations of CO in the eastern and northern suburb in Shijiazhuang were significantly higher
than those of urban areas (t-test, $p<0.01$). Note that the concentrations of $O_3$ in urban area and suburb
were lower during different stages of TECA (Fig. 5). Overall, during the TECA period, the effect of
control measures for atmospheric pollutants sources in suburb was better than in urban area,
especially for the effect of control measures for particulate matters sources. The effect of control
measures for CO was not notable during the TECA period, especially in suburb, likely due to the
increasing usage of domestic coal in suburb along with the temperature decreasing (Table 1).

----

**Fig. 5.** The spatial variations of atmospheric pollutants ($PM_{2.5}$, $PM_{10}$, $SO_2$, $NO_2$, CO and $O_3$) during the four stages
(NCANHP, NCAHP, CAHP and ACA) of the TECA period in Shijiazhuang. The pictures were produced by ArcGIS
based kriging interpolation method.

----





### 3.2 Variations of chemical species in PM$_{2.5}$


The average concentrations of chemical species in PM$_{2.5}$ in Shijiazhuang during the whole
sampling period are shown in Fig. 6. The annual mean concentrations of OC, SO$_4^{2-}$, NO$_3^-$ and NH$_4^+$
in PM$_{2.5}$ were 43.1 μg/m$^3$, 39.0 μg/m$^3$, 33.6 μg/m$^3$ and 25.6 μg/m$^3$, respectively, and their
contributions to PM$_{2.5}$ were up to 23.1 %, 20.0 %, 17.3 % and 12.3 %, respectively. The annual
mean concentrations of EC and Cl$^-$ were 11.7 μg/m$^3$ and 7.7 μg/m$^3$, respectively, which accounted
for 5.9 % and 4.1 % of PM$_{2.5}$. Note that the annual mean concentrations of elements in PM$_{2.5}$ were
relatively lower, which varied from 0.03 to 2.6 μg/m$^3$, accounting for 0.02-2.4 % of PM$_{2.5}$.
Compared to other elements, the annual mean concentrations of Si (2.6 μg/m$^3$) and Al (1.4 μg/m$^3$)
were relatively higher during the whole sampling period, which accounted for 2.4 % and 1.2 % of
PM$_{2.5}$, respectively. In this study, the annual mean concentrations of OC, SO$_4^{2-}$, NO$_3^-$ and NH$_4^+$ in
Shijiazhuang were clearly higher than Beijing (Gao et al., 2016), Tianjin (Wu et al., 2015), Jinan
(Gao et al., 2011), Shanghai (Wang et al., 2016c), Chengdu (Tao et al., 2013b), Xian (Wang et al.,
2015a), Hangzhou (Liu et al., 2015) and Heze (Liu et al., 2017a).
The values of P$_{i\text{-heating}}$ and P$_{i\text{-action}}$ of different chemical species in PM$_{2.5}$ were calculated by
using the Eq. (8) and (9). The variations of chemical species in PM$_{2.5}$ at four stages of the TECA
and the values of P$_{i\text{-heating}}$ and P$_{i\text{-action}}$ in Shijiazhuang are shown in Figs. 7 and 8. Compared to the
NCANHP, the concentrations of chemical species during the NCAHP showed a significantly
increased tendency (t-test, $p<0.01$), the concentrations of SO$_4^{2-}$, Cl$^-$, OC, EC, Si, Al, Ca$^{2+}$ and Mg$^{2+}$
increased by 7.9, 3.7, 6.7, 3.2, 1.6, 0.6, 0.4 and 0.1 μg/m$^3$, respectively, and the P$_{i\text{-heating}}$ values of
these species were up to 30.0 %, 40.2 %, 14.6 %, 22.1 %, 78.8 %, 63.5 %, 47.4 % and 45.9 %,
respectively, during the NCAHP. As these species (i.e., SO$_4^{2-}$, Cl$^-$, OC, EC, Si, Al, Ca$^{2+}$ and Mg$^{2+}$)
were closely associated with coal combustion (Cao et al., 2011; Liu et al., 2015; Liu et al., 2016;
Liu et al., 2017a, c), therefore, coal combustion for heating in winter probably had a great impact
on increasing of these chemical species in PM$_{2.5}$. Furthermore, compared to the NCANHP, the
concentrations of Cr, Cu, Fe, Mn, Ti, Zn and Pb increased by 0.02, 0.02, 0.34, 0.02, 0.02, 0.28 and
0.07 μg/m$^3$, respectively, and the P$_{i\text{-heating}}$ values of these species were 72.7 %, 33.1 %, 34.4 %,
21.0 %, 45.8 %, 48.3 % and 36.2 %, respectively, during the NCAHP. The Cr, Cu, Fe, Mn, Ti, Zn
and Pb were closely related to industrial sources (Liu et al., 2015; Kabala and Singh, 2001; Morishita
et al., 2011; Mansha et al., 2012; Yao et al., 2016), thus, the industrial emissions might have a higher



influence on PM$_{2.5}$ during the NCAHP than that during the NCANHP. Also, it might be closely
associated with the unfavorably meteorological factors (Table 1 and Fig. S7).

Compared to the NCAHP, the concentrations of SO$_4^{2-}$, Cl$^-$, OC and EC during the CAHP

increased by 16.8, 0.3, 19.8 and 14.6 μg/m$^3$, respectively, and the P$_{i\text{-action}}$ values of which were up
to -48.8 %, -2.0 %, -37.3 % and -83.0 %, respectively, during the CAHP. As coal combustion was
an important source of SO$_4^{2-}$, Cl$^-$, OC and EC (Cao et al., 2011; Liu et al., 2015; Liu et al., 2016;
Liu et al., 2017a, c), so it can be inferred that the influence of coal combustion might increase
apparently during the CAHP compared to the NCAHP, which was likely due to the increased usage
of the coal for domestic heating with the reduction of temperature during winter (Table 1). As also
Fig. 5 shown that the concentrations of CO during the CAHP were higher than those during the
NCAHP, especially in rural areas. Furthermore, OC and EC were associated with the vehicle exhaust
(Liu et al., 2016; Liu et al., 2017a), thus, the effect of motor vehicle management and control
measures during the CAHP might be offset by the unfavorably meteorological conditions to some
extent during the CAHP (Table 1 and Fig. S7). However, compared to the NCAHP, the
concentrations of Si, Al, Ca$^{2+}$ and Mg$^{2+}$ during the CAHP decreased by 1.1, 0.1, 0.6 and 0.1 μg/m$^3$,
respectively, and the P$_{i\text{-action}}$ values of which were up to 30.3 %, 4.5 %, 47.0 % and 45.2 %,
respectively. As Si, Al, Ca$^{2+}$ and Mg$^{2+}$ were mainly originated from the crustal dust (Liu et al., 2016;
Shen et al., 2010; Wang et al., 2015a; Yang et al., 2016b), therefore, the influence of crustal dust on
PM$_{2.5}$ during the CAHP might decrease clearly compared to the NCAHP. That's closely related to
the control measures of inhabitation of dust emission during the TECA period (as shown in section
2.5). In general, from the view of the variation of PM$_{2.5}$ speciation, there was no doubt that the
TECA had a certain positive environmental effect on the improvement of air quality. However, the
ambient pollutant concentration was impacted by not only the emission sources, but also the
meteorological conditions, regional background level and distant transportation, it was
understandable that the concentration of CO had a "rebound" effect during the CAHP as the height
of mixing layer was only 474 m and a low wind speed of 0.4 m/s.

----

**Fig. 6.** The average concentrations and percentages of chemical species in PM$_{2.5}$ in Shijiazhuang during the whole
sampling period: November 24, 2015 to January 9, 2017.
**Fig. 7.** The variations of chemical species in PM$_{2.5}$ during the four stages (NCANHP, NCAHP, CAHP and ACA) of
the TECA period.



**Fig. 8.** The $P_{i\text{-heating}}$ and $P_{i\text{-action}}$ of chemical species in $PM_{2.5}$ during the TECA period in Shijiazhuang.

----

### 3.3 Variations of PM2.5 sources contributions

The filter membrane samples of $PM_{2.5}$ were collected in three sites (LQ, LC and TSMS) in Shijiazhuang from November 24, 2015 to January 9, 2017, and source apportionment was carried out by using EPA PMF 5.0, as well as five factors were identified during the period (Figs. 9 and 10). The chemical profile of factor 1 was mainly represented by Si (72.3 %), $Ca^{2+}$ (74.0 %), $Mg^{2+}$ (43.9 %) and Al (71.3 %), which were derived mainly from crustal dust (Liu et al., 2016; Shen et al., 2010; Wang et al., 2015a). Thus, factor 1 was viewed as crustal dust. The contribution proportions of factor 1 to $PM_{2.5}$ decreased from 19.5 % (38.5 μg/m³) during the whole year (WY) to 15.0 % (40.3 μg/m³) during the CAHP, and increased up to 16.3 % (48.3 μg/m³) during the ACA. The main species of factor 2 were $SO_4^{2-}$ (53.9 %), $NO_3^-$ (89.8 %) and $NH_4^+$ (75.0 %). Therefore, it was easily identified as secondary sources (Liu et al., 2015, 2016, 2017a; Santacatalina et al., 2010; Srimuruganandam and Nagendra, 2012). The contribution proportions of factor 2 to $PM_{2.5}$ ranged from 32.7 % (64.6 μg/m³) during the WY to 31.6 % (84.8 μg/m³), and decreased to 28.8 % (85.2 μg/m³) during the ACA. Factor 3 was represented by the relatively high loadings of OC (55.9 %), EC (70.9 %), Cu (26.9 %) and Zn (26.5 %). Given that the OC and EC are generally predominant in the reported source profile of vehicle exhaust (Liu et al., 2016, 2017a; Yao et al., 2016), and Zn is widely used as an additive for lubricant in two-stroke engines, and Cu is closely associated with brake wear (Begum et al., 2004; Canha et al., 2012; Lin et al., 2015; Liu et al., 2017a). Therefore, factor 3 was identified as vehicle emissions. The contribution proportions of factor 3 to $PM_{2.5}$ decreased from 13.4 % (26.4 μg/m³) during the WY to 10.6 % (28.5 μg/m³) during the CAHP, and increased to 14.1 % (41.7 μg/m³) during the ACA. Factor 4 was characterized by the high contributions of $Ca^{2+}$ (26.0 %), $Mg^{2+}$ (31.0 %), Si (13.3 %), As (84.9 %), $Cl^-$ (38.6 %), OC (20.2 %) and $SO_4^{2-}$ (26.7 %), and the combination of these species in factor 4 inferred they were co-emission from coal combustion (Cao et al., 2011; Liu et al., 2015, 2016, 2017a,c; Zhang et al., 2011). Therefore, factor 4 was identified as coal combustion. The contribution proportions of factor 4 to $PM_{2.5}$ increased from 26.2 % (51.7 μg/m³) during the WY to 31.7 % (85.2 μg/m³) during the CAHP, and lightly increased to 32.6 % (96.3 μg/m³) during the ACA. Factor 5 was identified as industrial emissions, with high loadings of Cr (66.7 %), Cu (63.7 %), Fe (83.2 %), Mn (51.3 %), Ti (70.0 %), Zn (69.2 %), Pb (42.1 %) and $Cl^-$





(41.0 %) (Almeida et al., 2015; Liu et al., 2015, 2016; Morishita et al., 2011; Mansha et al., 2012;
Yao et al., 2016). The contribution proportions of factor 5 to $PM_{2.5}$ ranged from 5.1 % (10.0 μg/m$^3$)
during the WY to 5.3 % (14.2 μg/m$^3$) during the CAHP, and decreased to 4.9 % (14.4 μg/m$^3$) during
the ACA. Note that the contribution of industrial emissions to $PM_{2.5}$ was relatively lower than other
sources (Fig. 10).
In general, crustal dust, secondary sources, vehicle emissions, coal combustion and industrial
emissions were identified as $PM_{2.5}$ sources in Shijiazhuang. Compared to the WY, the contribution
of coal combustion to $PM_{2.5}$ increased significantly during the CAHP and the ACA, which was
closely associated with the coal heating for cold winter (Liu et al., 2016), and the unfavorably
meteorological conditions (Table 1 and Fig. S7). Compared to the WY, the contribution proportions
of crustal dust and vehicle emissions to $PM_{2.5}$ decreased apparently during the CAHP; and compared
to the ACA, the contribution concentrations and proportions of which to $PM_{2.5}$ also decreased
significantly during the CAHP. It indicated that the control effects of motor vehicles and crustal dust
were remarkable during the CAHP, and the results were consistent with the above analysis. The
contribution proportions of secondary sources to $PM_{2.5}$ during the CAHP and ACA showed little
change compared to the WY. However, the contribution concentrations of secondary sources to
$PM_{2.5}$ increased significantly during the CAHP and the ACA compared to the WY, likely due to
frequent hazy events during the period, when there were significant secondary reactions (Han et al.,
2014; Li et al., 2016a). In addition, it also illustrated that the discharge of atmospheric pollutants
was still enormous even under such strict control measures. Note that the contribution
concentrations of industrial emissions to $PM_{2.5}$ during the CAHP and ACA increased apparently
compared to the WY, which probably was affected by adverse weather conditions during the period
(Table 1 and Fig. S7).
Chen et al. (2016b) reported that the concentrations of particles during the 2014 Youth Olympic
Games (YOG) period (August) were much lower than before-Games period (July) and after-Games
period (September); and fugitive dusts, construction dusts and secondary sulfate aerosol decreased
obviously in YOG, which means mitigation measures have played an effective role in reduction of
particulate matter. Wang et al. (2017) found that the contributions of vehicles, industrial sources,
fugitive dust, and other sources decreased 13.5-14.7 %, 10.7-11.2 %, 4.5-5.6 % and 1.7-2.7 %,
respectively, during the Asia-Pacific Economic Cooperation (APEC) and China's Grand Military





Parade (CGMP), compared to the period before the control actions. Guo et al. (2013) found that
primary vehicle contributions were reduced by 30 % at the urban site and 24 % at the rural site,
compared with the non-controlled period before the Beijing 2008 Olympics. The reductions in coal
combustion contributions were 57 % at PKU site and 7 % at Yufa site. As we can see that these
control-actions of the strict measures taken for emission sources during the international events held
in China, including the TECA in Shijiazhuang, were all very important practical exercises and rarely
scientific experiments. However, it cannot be advocated as the normalized control measures for
atmospheric pollution in China. These strict measures taken during these periods are temporary, and
there is a normal recovery of all the emissions of sources after the operation. Once adverse weather
conditions occur, and the hazy events may continue to happen eventually. In short, the direct cause
of the severe atmospheric pollution in China is that the emission of pollutants beyond the air
environment's self-purification capacity, and the essential reason is unreasonable and unhealthy
pattern for economic development of China.

----

**Fig. 9.** Source profiles obtained with the PMF for $PM_{2.5}$. Filled bars identify the species that mainly characterize
each factor profile.
**Fig. 10.** Source contributions of $PM_{2.5}$ during different stages in Shijiazhuang. WY represents whole year: November
24, 2015 to January 9, 2017.

----

**3.4 Backward trajectory and PSCF analysis**
The backward trajectory analysis was used to identify the transport pathways of the air mass
during the CAHP. In terms of the directions and travelled areas, these trajectories were divided into
the five groups (Fig. 11). Trajectory clusters 1, accounting for 31.3 % of the total, originated from
Shanxi province and passed over North of Hebei before arriving at Shijiazhuang. Trajectory cluster
1 reflected the features of small-scale, short-distance air mass transport (Fig. 11). The higher
concentrations of $PM_{10}$ (358 μg/m$^3$), $PM_{2.5}$ (237 μg/m$^3$) and CO (3.9 μg/m$^3$) might be due to the
variety of emission sources and the accumulation of pollutants from surrounding areas, since the
moving speed of air mass in cluster 1 was much lower than other trajectories (Fig. 11 and Table 2).
Trajectory cluster 2, 3 and 4 accounted for 58.0 % of the total trajectories, and began from the
northwest of China, passed through the Inner Mongolia and Shanxi, showing the features of large-
scale, long-distance air transports. The relative lower concentrations of $PM_{10}$ (189-290 μg/m$^3$),



PM$_{2.5}$ (119-181 µg/m$^3$), SO$_2$ (50-67 µg/m$^3$), NO$_2$ (58-78 µg/m$^3$) and CO (2.1-3.0 mg/m$^3$) were
closely associated with high moving speeds of air mass (Fig. 11 and Table 2), and relatively less
anthropogenic emission sources in the northwest of China. Trajectory cluster 5 was mainly
originated from Ningxia province, passed over Shaanxi, Shanxi and Hebei before arriving at
Shijiazhuang, accounting for 10.8 % of the total, showing the features of small-scale, short-distance
air transport significantly elevated levels of PM$_{10}$ (451 µg/m$^3$), PM$_{2.5}$ (303 µg/m$^3$), SO$_2$ (83 µg/m$^3$),
NO$_2$ (104 µg/m$^3$) and CO (4.8 mg/m$^3$) with trajectory cluster 5 were associated with the sources and
the accumulation of pollutants from surrounding areas. As well known that the Beijing-Tianjin-
Hebei region was one of the severest polluted areas in China (Bi et al., 2014; Chen et al., 2013; Gu
et al., 2011; Wang et al., 2014; Zhao et al., 2012), it might be an important reason why the
concentrations of pollutants were higher with trajectory clusters 1 and 5 (Fig. 11 and Table 2).

In this study, PSCF model was used to analyze the potential sources-areas of atmospheric

pollutants by combining backward trajectories and the concentrations of atmospheric pollutants in
Shijiazhuang during the CAHP, and the results were shown in Fig. 12. The values of weighted
potential source contribution function (WPSCF) of CO were higher in the north of Shaanxi, south
of Shanxi and central and southern Inner Mongolia, which were mainly potential sources-areas of
CO concentrations in Shijiazhuang (Fig. 12 (a)). The WPSCF values of NO$_2$ were higher in north
of Henan and Shaanxi, Hebei, Shanxi, and central and southern Inner Mongolia, which were mainly
potential sources-areas of NO$_2$ concentrations in Shijiazhuang (Fig. 12 (b)). The WPSCF values of
O$_3$ and SO$_2$ were higher in the north of Henan and Shaanxi, Shanxi, and south of Hebei, which were
distinguished as major potential sources-areas of O$_3$ and SO$_2$ concentrations in Shijiazhuang (Fig.
12 (c) and (d)). Moreover, the southwest of Shandong was also identified as mainly potential
sources-areas of SO$_2$ concentrations in Shijiazhuang. As for PM$_{2.5}$ and PM$_{10}$, the WPSCF values
were higher in south of Hebei, and east of Shanxi, which were identified as mainly potential sources-
areas of PM$_{2.5}$ and PM$_{10}$ concentrations in Shijiazhuang (Fig. 12 (e) and (f)). Overall, the potential
sources-areas of the atmospheric pollutants in Shijiazhuang mainly concentrated in the surrounding
regions of Shijiazhuang, including south of Hebei, north of Henan and Shanxi. Previous studies also
reported that Shanxi, Hebei and Henan provinces had serious air pollution problems (Feng et al.,
2016; Kong et al., 2013; Meng et al., 2016; Zhu et al., 2011), revealing the regional nature of the
atmospheric pollution in Northern Plain of China. Therefore, there is an urgent need for making



cross-boundary control policy except for local control-measures given the high background level of
pollutants.

----

**Fig. 11.** Five clusters of the 72-h air mass backward trajectories during the CAHP. Red star represents Shijiazhuang
city.
**Fig. 12.** Potential sources areas of atmospheric pollutants obtained from PSCF model during the CAHP. Red star
represents Shijiazhuang city. The colors represent potential sources-areas influenced on the atmospheric pollutants,
and the red color could be determined to be relatively important sources-areas while the blue color means
unimportant potential sources-areas.
**Table.2** The average concentrations of atmospheric pollutants in different clusters during the CAHP.

----

### 4 Conclusions


The control measures of atmospheric pollution in Shijiazhuang were effective and was in a right
direction. Under unfavorably meteorological conditions, the mean concentrations of $PM_{2.5}$, $PM_{10}$,
$SO_2$, $NO_2$, and chemical species (Si, Al, $Ca^{2+}$, $Mg^{2+}$) in $PM_{2.5}$ during the CAHP significantly
decreased compared to the NCAHP. Overall, the effects of control measures in suburbs were better
than in urban area, especially for the effects of control measures for particulate matters sources. The
effects of control measures for CO emission sources were not apparent during the CAHP, especially
in suburbs.
The pollutant's emission sources during the CAHP were in effective control, especially for
crustal dust and vehicles. While the necessary coal heating for cold winter and the unfavorable
meteorological conditions had an offset effect on the control measures for emission sources to some
extent. The discharge of pollutants was still enormous even under such strict control measures.
The backward trajectory and PSCF analysis in the light of atmospheric pollutants suggested
that the potential sources-areas mainly concentrated in surrounding regions of Shijiazhuang, i.e.,
south of Hebei, north of Henan and Shanxi. The regional nature of the atmospheric pollution in
Northern China Plain revealed that there is an urgent need for making cross-boundary control policy
except for local control-measures given the high background level of pollutants.
The TECA is an important practical exercise but it can't be advocated as the normalized control
measures for atmospheric pollution in China. The direct cause of atmospheric pollution in China is
the emission of pollutants exceeds the air environment's self-purification capacity, and the essential
reason is unreasonable and unhealthy pattern for economic development of China.





**Acknowledgments**

This study was financially supported by the National Key Research and Development

Program of China (2016YFC0208500 & 2016YFC0208501) and Tianjin Science and Technology
Foundation (16YFZCSF00260) and the National Natural Science Foundation of China (21407081)
and the Fundamental Research Funds for the Central Universities. The authors thank Shijiazhuang
Environmental Protection Monitoring Station for their participating in the sampling campaign and
chemical analysis of samples.

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



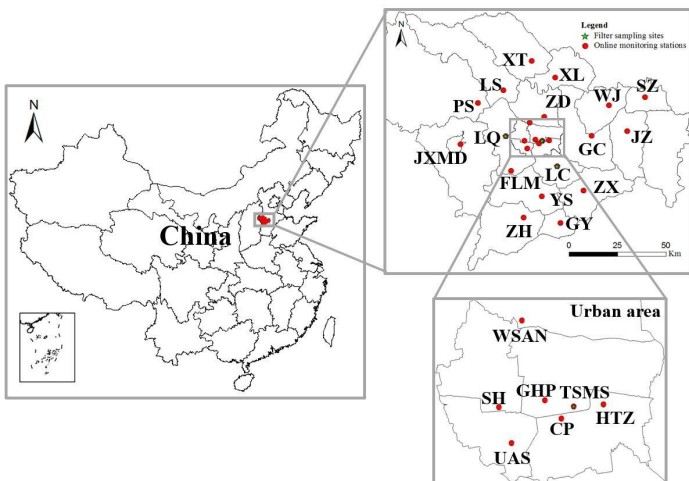

**Fig. 1.** Maps of the online monitoring stations and the filter membrane sampling sites in Shijiazhuang. The 24 online monitoring stations mainly include Twenty-second Middle School (TSMS), Fenglong Mountain (FLM), High-tech Zone (HTZ), Great Hall of the People (GHP), Century Park (CP), Water Source Area in the Northwest (WSAN), University Area in the Southwest (UAS), Staff Hospital (SH), Gaoyi (GY), Gaocheng (GC), Xingtang (XT), Jinzhou (JZ), Jingxing Mining District (JXMD), Lingshou (LS), Luquan (LQ), Luancheng (LC), Pingshan (PS), Shenze (SZ), Wuji (WJ), Xinle (XL), Yuanshi (YS), Zanhuang (ZH), Zhaoxian (ZX) and Zhengding (ZD). The filter membrane sampling sites are mainly located in TSMS, LQ and LC.

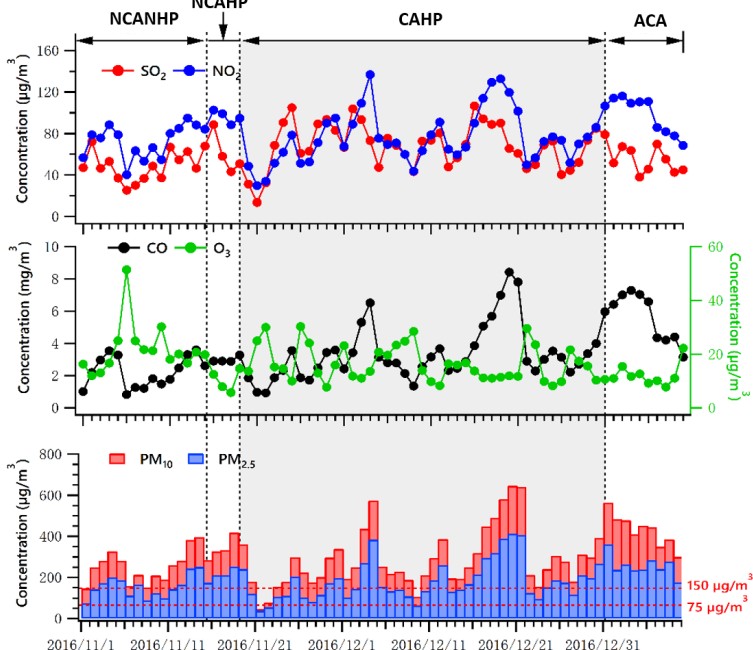

**Fig. 2.** The variations of atmospheric pollutants concentrations during the four stages (NCANHP, NCAHP, CAHP and ACA) of the TECA period in Shijiazhuang.



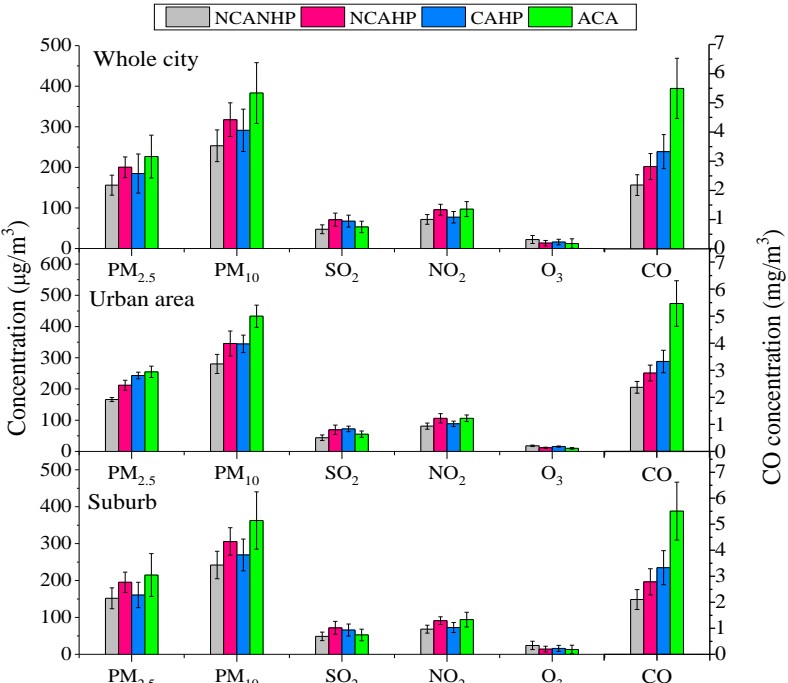

**Fig. 3.** The concentrations variations of PM$_{2.5}$, PM$_{10}$ and gaseous pollutants during the four stages (NCANHP, NCAHP, CAHP and ACA) of the TECA period in Shijiazhuang.

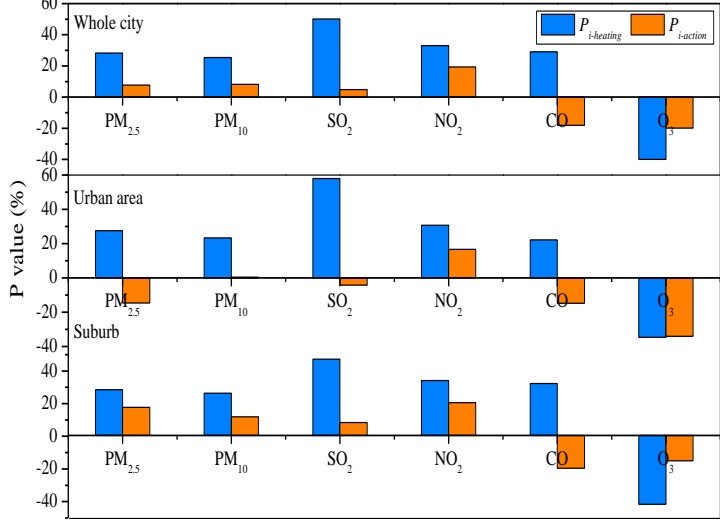

**Fig. 4.** The P$_{i-heating}$ and P$_{i-action}$ of PM$_{2.5}$, PM$_{10}$ and gaseous pollutants (SO$_2$, NO$_2$, CO and O$_3$) calculated by equation (8) and (9) in urban area and suburb in Shijiazhuang.





**Fig. 5.** The spatial variations of atmospheric pollutants (PM2.5, PM10, SO2, NO2, CO and O3) during the four stages (NCANHP, NCAHP, CAHP and ACA) of the TECA period in Shijiazhuang. The pictures were produced by ArcGIS based kriging interpolation method.





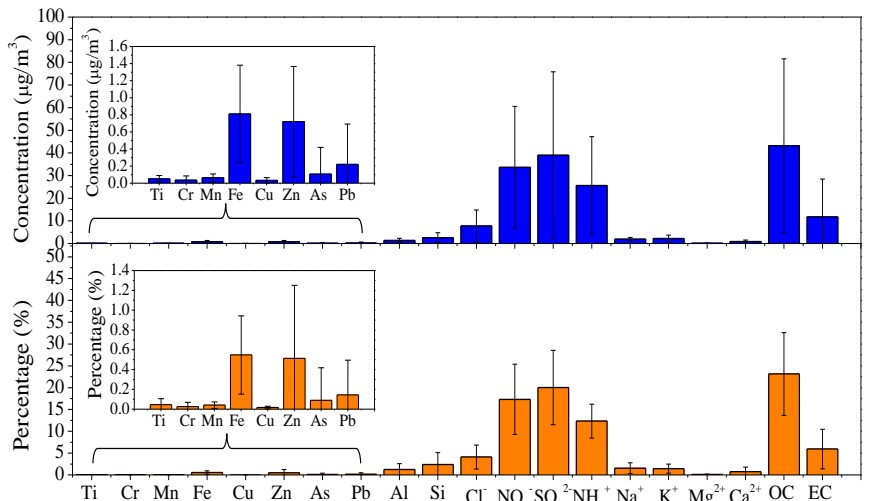

**Fig. 6.** The average concentrations and percentages of chemical species in PM$_{2.5}$ in Shijiazhuang during the whole sampling period: November 24, 2015 to January 9, 2017.

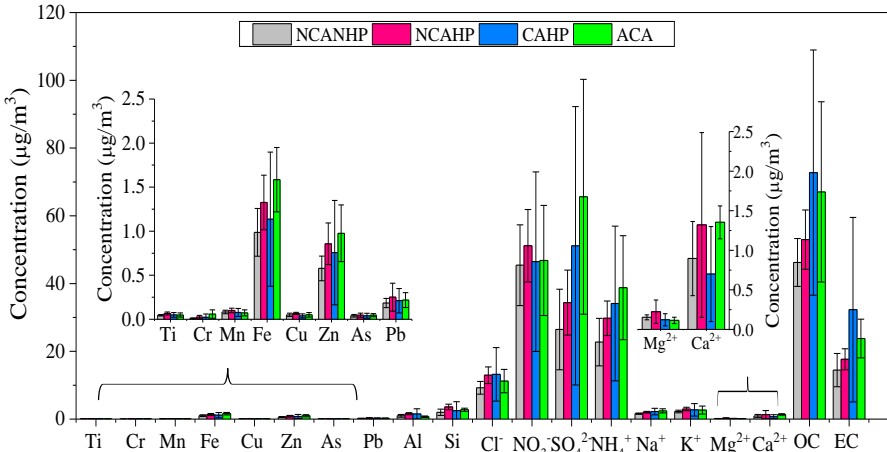

**Fig. 7.** The variations of chemical species in PM$_{2.5}$ during the four stages (NCANHP, NCAHP, CAHP and ACA) of the TECA period.



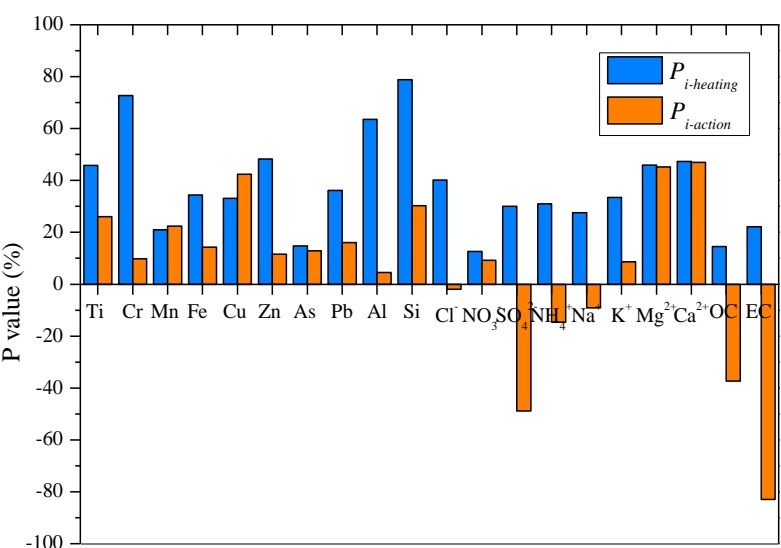

**Fig. 8.** The $P_{i\text{-heating}}$ and $P_{i\text{-action}}$ of chemical species in $PM_{2.5}$ during the TECA period in Shijiazhuang.

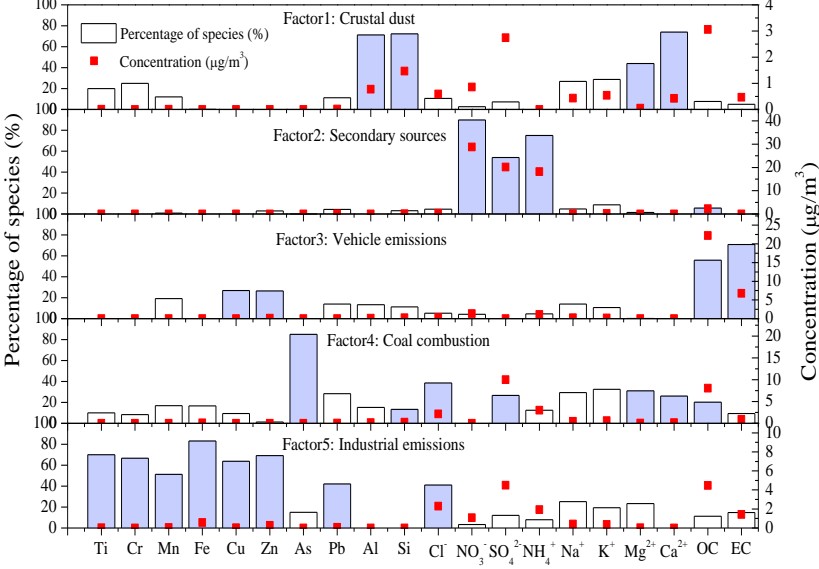

**Fig. 9.** Source profiles obtained with the PMF for $PM_{2.5}$. Filled bars identify the species that mainly characterize each factor profile.




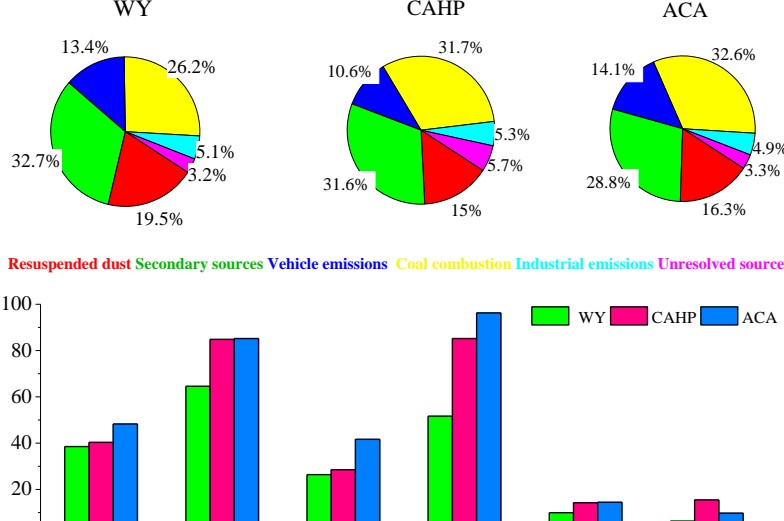

**Fig. 10.** Source contributions of PM$_{2.5}$ during different stages in Shijiazhuang. WY represents whole year: November 24, 2015 to January 9, 2017.

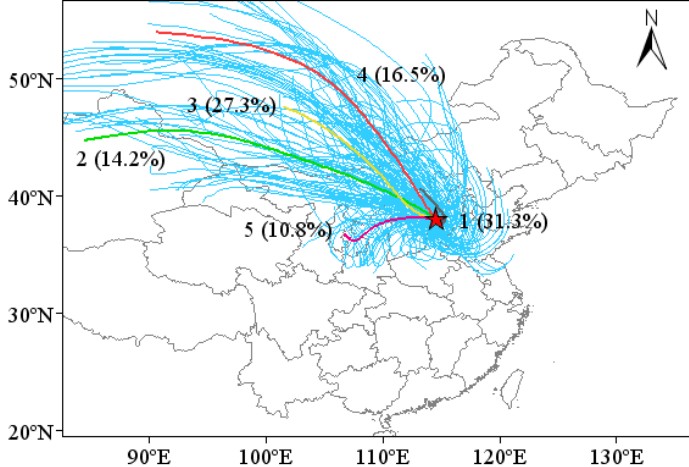

**Fig. 11.** Five clusters of the 72-h air mass backward trajectories during the CAHP. Red star represents Shijiazhuang city.



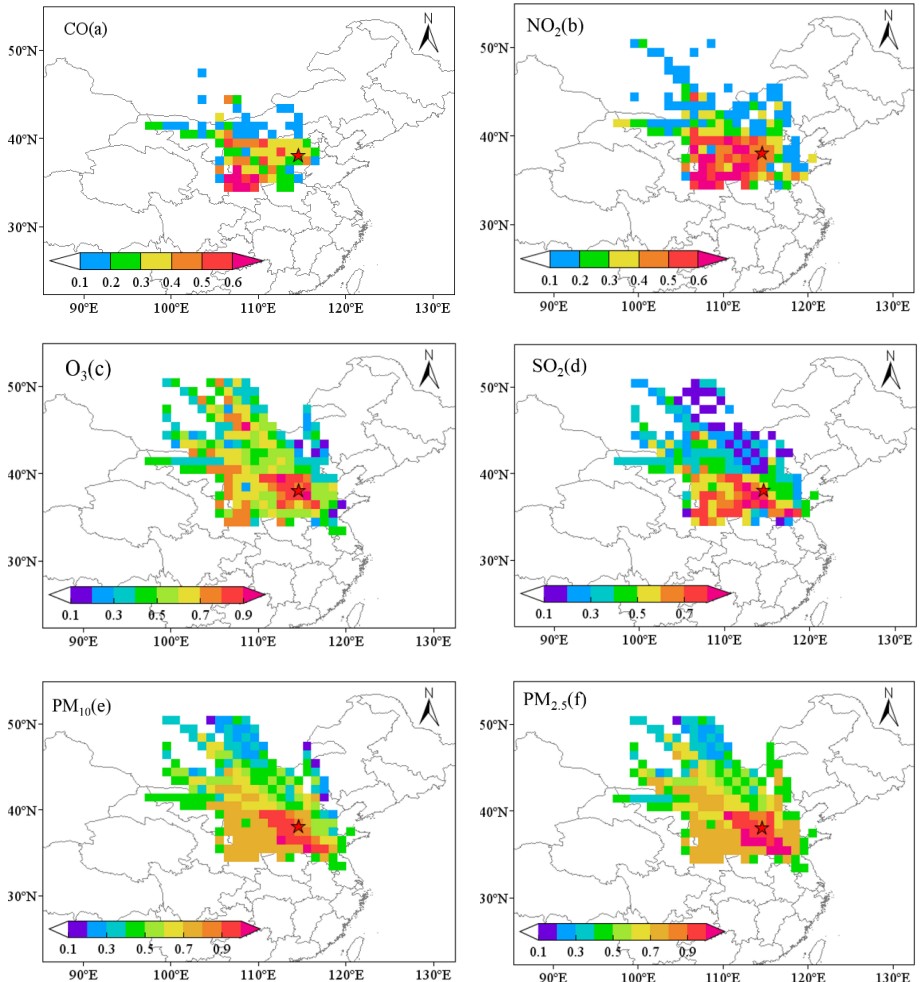

**Fig. 12.** Potential sources areas of atmospheric pollutants obtained from PSCF model during the CAHP. Red star represents Shijiazhuang city. The colors represent potential sources-areas influenced on the atmospheric pollutants, and the red color could be determined to be relatively important sources-areas while the blue color means unimportant potential sources-areas.



**Table 1.** The meteorological conditions during the four stages (NCANHP, NCAHP, CAHP and ACA) of the TECA period in Shijiazhuang.

|  | NCANHP | | NCAHP | | CAHP | | ACA | |
|---|---|---|---|---|---|---|---|---|
|  | Ave. | S.D. | Ave. | S.D. | Ave. | S.D. | Ave. | S.D. |
| Temperature (℃) | 8.4 | 3.6 | 7.4 | 2.4 | 3.1 | 3.8 | 0.7 | 2.7 |
| Relative humidity (%) | 77.7 | 17.0 | 73.4 | 15.7 | 71.5 | 18.0 | 83.3 | 18.1 |
| Wind speed (m/s) | 0.7 | 1.2 | 0.6 | 0.6 | 0.4 | 1.0 | 0.5 | 1.1 |
| Height of mixed layer (m) | 540 | 144 | 590 | 274 | 474 | 299 | 431 | 360 |

Ave. represents average value, S.D. represents standard deviation. NCANHP represents the no control action and no heating period, NCAHP represents the no control action and heating period, CAHP represents the control action and heating period, and ACA represents after control action.

**Table 2.** The average concentrations of atmospheric pollutants in different clusters during the CAHP.

| Clusters | Probability of occurrence (%) | Atmospheric Pollutants ($\mu g/m^3$) | | | | | |
|---|---|---|---|---|---|---|---|
|  |  | $SO_2$ | $NO_2$ | $O_3$ | $CO(mg/m^3)$ | $PM_{10}$ | $PM_{2.5}$ |
| 1 | 31.3 | 68 | 88 | 14 | 3.9 | 358 | 237 |
| 2 | 14.2 | 67 | 78 | 24 | 3.0 | 290 | 181 |
| 3 | 27.3 | 65 | 69 | 20 | 2.8 | 232 | 152 |
| 4 | 16.5 | 50 | 58 | 27 | 2.1 | 189 | 119 |
| 5 | 10.8 | 83 | 104 | 16 | 4.8 | 451 | 303 |