# Peer review of "Effectiveness evaluation of temporary emission control action in 2016 winter in Shijiazhuang, China"

_Atmospheric Chemistry and Physics, 2017_

## Referee Comment (RC1) · Anonymous Referee #1 · 15 Jan 2018

This manuscript reports a study that evaluates the effectiveness of temporary emissions controls during 2016 winter in Shijiazhuang, China by utilizing measurements of standard air pollutants' concentrations and filter measurements of concentrations of PM2.5 and its components. The entire study period was divided into four sub-periods: NCANHP, NCAHP, CAHP, ACA. By defining P-heating and P-action as differences in concentrations measured during the certain sub-periods, the authors conclude on the effects of heating and emission controls on the local air quality. The authors also employed PMF for source contribution analyses and conducted backward trajectory and PSCF analysis. Several concerns came from this reviewer: (1) NCAHP and ACA, both are a sub-period of no control plus heating period by nature, why treat the two

sub-periods differently and only include NCAHP in P-heating and P-action definition. Considering NCAHP is only 3 days by definition, and it is at the very start of the heating season, isn't NCAHP a special period against the entire heating season? (2) In PMF results section, there is inconsistency from the previous analyses to report source contribution results only for CAHP and ACA, why left out NCANHP and NCAHP for analysis in source contribution changes? This is where the observations can be somehow directly traced back to the control strategies, it needs a better usage of the materials. (3) PSCF analysis itself is good to weigh relative importance of transported source impacts. However, does the PSCF results help much here to add anything on making the major conclusion, especially on the condition that it doesn't tell anything about the relative importance between the local and transport contributions?

---

## Referee Comment (RC2) · Anonymous Referee #2 · 11 Feb 2018

The authors make an assessment of emission controls during the heating season in one of the most polluted cities in China. They performed field studies in Shijiazhuang for two months, analyzed ions and trace elements in PM2.5, and used PMF and HYS-PLIT model to investigate the sources and evolution processes of air pollution in and around the city. This paper involves lots of work, and I find some results very interesting, for example, the improvement of air quality due to emission controls is more evident in suburbs than urban area. I can believe the emission control measures might have considerable effectiveness in improving air quality, but I have doubts regarding the method of quantification this paper used, which does not look convincing to me. I don't see much scientific significance in this paper in its current form though it summarized

plenty of data and did some analysis. The paper is not well written and needs lots of editing. My major concern is as follows:

1. The authors have found the height of mixed layer and wind speed have a significant influence on air pollutant concentrations, but they use an oversimplified method (Eqs. 8 and 9) to quantify the impact of a single variable (i.e., heating and emission controls) on air pollution, without excluding the effects of meteorological conditions quantitatively. The method does not make sense.

2. The error bars in Fig. 3 are too long, and thus the average values are uncertain. Some of the error bars are even longer than the reduction of concentrations caused by emissions controls (calculated by Eq. 9). For example, the error bar of PM2.5 concentrations during the period of CAHP (blue bar) in the whole city is much larger than the reductions compared to the NCAHP period (pink bar). There are many such cases in Fig. 3, as well as in Fig. 7, which makes the quantification analysis based on these data look not convincing.

3. Some statement in the main text are very misleading. For example, in lines 366–367, "Well known that the NO2 is mainly derived from the vehicle exhaust (Liu et al., 2017b). Therefore, the control effect of motor vehicles was remarkable during the CAHP in Shijiazhuang." And in lines 391–392, the authors say "Furthermore, the effects of control measures for domestic coal might be worse during the CAHP." I don't think the data and analysis this paper presents can lead to such conclusions. The authors should be more precise what they find from their study.
* * *

---

## Author Comment (AC1) · 22 Mar 2018

This manuscript reports a study that evaluates the effectiveness of temporary emissions controls during 2016 winter in Shijiazhuang, China by utilizing measurements of standard air pollutants' concentrations and filter measurements of concentrations of $PM_{2.5}$ and its components. The entire study period was divided into four sub-periods: NCANHP, NCAHP, CAHP, ACA. By defining P-heating and P-action as differences in concentrations measured during the certain sub-periods, the authors conclude on the effects of heating and emission controls on the local air quality. The authors also employed PMF for source contribution analyses and conducted backward trajectory and

[Figure]

PSCF analysis. Several concerns came from this reviewer: (1) NCAHP and ACA, both are a sub-period of no control plus heating period by nature, why treat the two sub-periods differently and only include NCAHP in P-heating and P-action definition. Considering NCAHP is only 3 days by definition, and it is at the very start of the heating season, isn't NCAHP a special period against the entire heating season?

Response: Yes, NCAHP and ACA, both are a sub-period of no control plus heating period by nature. However, the meteorological conditions during the NCANHP and ACA were significantly distinct (Table 1). In addition, from NCANHP to ACA, with the decrease of temperature (Table 1), the load of coal-fired heating is different, and the emission and intensity of pollutants from coal combustion are significantly different. Therefore, the NCANHP and ACA are separately analyzed in this study. The related contents have been revised in the revised manuscript (on the line 336). Indeed, the NCANHP is the beginning of heating period, and the load and degree of coal-fired heating are lower compared to other stages, so that the representative of the heating period for NCANHP is limited. However, the main purpose of this study is to evaluate the effectiveness of temporary emission control action. Compared to the beginning of the heating period, the control effects of temporary emission control action is found to be obvious (as shown in the manuscript); therefore, compared to the other stage during which the heating loads are higher, the control effects of temporary emission control action should be more obvious. Therefore, it is possible to use NCAHP to evaluate the effects of control action. In addition, the weather conditions in the ACA are significantly worse than the CAHP (Table 1). If ACA is used to calculate P-heating for evaluating the effects of control action, the effect of control action is not well evaluated.

(2) In PMF results section, there is inconsistency from the previous analyses to report source contribution results only for CAHP and ACA, why left out NCANHP and NCAHP for analysis in source contribution changes? This is where the observations can be somehow directly traced back to the control strategies, it needs a better usage of the materials.

Response: We have added the results of source apportionment during the NCANHP and NCAHP to the revised manuscript (Fig. 10). The related contents in the revised manuscript have been revised based on the added contents.

(3) PSCF analysis itself is good to weigh relative importance of transported source impacts. However, does the PSCF results help much here to add anything on making the major conclusion, especially on the condition that it doesn't tell anything about the relative importance between the local and transport contributions?

Response: PSCF model can only qualitatively analyze the potential source-areas that affect the concentrations of air pollutants in Shijiazhuang (Lucey et al., 2001; Liu et al., 2017; Zhang et al., 2017; Zong et al., 2018), which cannot be quantified by local source and regional transmission. The purpose of this study is to evaluate the effects of control measures for local sources. Although quantitative research on regional contribution is also important, it is not the focus of the work. We will follow up the relevant research. In view of the length of this paper and the aim of the study, this article is not easy to analyze in details.

Lucey, D., Hadjiiski, L., Hopke, P.K., Scudlark, J.R., Church, T.: Identification of sources of pollutants in precipitation measured at the mid-Atlantic US coast using potential source contribution function (PSCF), Atmos. Environ., 35, 3979–3986, 2001. Liu, B.S., Wu, J.H., Zhang, J.Y., Wang, L., Yang, J.M., Liang, D.N., Dai, Q.L., Bi, X.H., Feng, Y.C., Zhang, Y.F., Zhang, Q.X.: Characterization and source apportionment of PM2.5 based on error estimation from EPA PMF5.0 model at a medium city in China, Environ. Pollut., 222, 10–22, 2017. Zhang, Y., Zhang, H., Deng, J., Du, W., Hong, Y., Xu, L., Qiu, Y., Hong, Z., Wu, X., Ma, Q., Yao, J., Chen, J.: Source regions and transport pathways of PM2.5 at a regional background site in East China, Atmos. Environ., 167, 202–211, 2017. Zong, Z., Wang, X.P., Tian, C.G., Chen, Y.J., Fu, S.F., Qu, L., Ji, L., Li, J., Zhang, G.: PMF and PSCF based source apportionment of PM2.5 at a regional background site in North China, Atmos. Res., 203, 207–215, 2018.

Please also note the supplement to this comment:
https://www.atmos-chem-phys-discuss.net/acp-2017-1001/acp-2017-1001-AC1-supplement.zip

[Figure]

[Figure]

[Figure]

Fig. 10. Source contributions of PM$_{2.5}$ during different stages in Shijiazhuang. WY represents whole year: November 24, 2015 to January 9, 2017.

Fig. 1.

**Table 1.** The meteorological conditions during the four stages (NCANHP, NCAHP, CAHP and ACA) of the TECA period in Shijiazhuang.

|  | NCANHP | | NCAHP | | CAHP | | ACA | |
|---|---|---|---|---|---|---|---|---|
|  | Ave. | S.D. | Ave. | S.D. | Ave. | S.D. | Ave. | S.D. |
| Temperature (℃) | 8.4 | 3.6 | 7.4 | 2.4 | 3.1 | 3.8 | 0.7 | 2.7 |
| Relative humidity (%) | 77.7 | 17.0 | 73.4 | 15.7 | 71.5 | 18.0 | 83.3 | 18.1 |
| Wind speed (m/s) | 0.7 | 1.2 | 0.6 | 0.6 | 0.4 | 1.0 | 0.5 | 1.1 |
| Height of mixed layer (m) | 540 | 144 | 590 | 274 | 474 | 299 | 431 | 360 |

Ave. represents average value, S.D. represents standard deviation. NCANHP represents the no control action and no heating period, NCAHP represents the no control action and heating period, CAHP represents the control action and heating period, and ACA represents after control action.

**Fig. 2.**

---

## Author Comment (AC2) · 22 Mar 2018

The authors make an assessment of emission controls during the heating season in one of the most polluted cities in China. They performed field studies in Shijiazhuang for two months, analyzed ions and trace elements in PM2.5, and used PMF and HYS-PLIT model to investigate the sources and evolution processes of air pollution in and around the city. This paper involves lots of work, and I find some results very interesting, for example, the improvement of air quality due to emission controls is more evident in suburbs than urban area. I can believe the emission control measures might have considerable effectiveness in improving air quality, but I have doubts regarding the

method of quantification this paper used, which does not look convincing to me. I don't see much scientific significance in this paper in its current form though it summarized plenty of data and did some analysis. The paper is not well written and needs lots of editing. My major concern is as follows: 1. The authors have found the height of mixed layer and wind speed have a significant influence on air pollutant concentrations, but they use an oversimplified method (Eqs. 8 and 9) to quantify the impact of a single variable (i.e., heating and emission controls) on air pollution, without excluding the effects of meteorological conditions quantitatively. The method does not make sense.

Response: We agree with the views of the reviewer. Actually, Pi-heating from Eq. (8) represents the increasing percentage (%) of atmospheric pollutant concentration because of the combined effects of heating for cold winter and meteorological conditions. Pi-action from Eq. (9) represents the decreasing percentage (%) of atmospheric pollutant concentration because of the combined influences of control action and meteorological conditions. The related contents have been added to the revised manuscript (on the line 350-352).

In this study, we focus on the calculation results from Eq. (9). The mean height of mixed layer, the mean wind speed and temperature during the CAHP were lower than those during the NCAHP (Table 1). Unfavorably meteorological conditions during the CAHP had an offset effect on the control measures for emission sources. In view of Eq. (9), it can be seen that the positive values for Pi-action were more able to show that control action was effective. If the values of Pi-action were negative, it shows that the control effect is not obvious or unfavorably meteorological conditions counteract the control effects. The related contents have been added to the revised manuscript (on the line 363-367).

Here, we have only carried out qualitative analysis to the meteorological conditions. However, for a short term process, there are no reliable quantitative methods based on the related literature researches over the world for model simulation. The main reason is that the simulation results of boundary layer and meteorology from the air quality

model during heavy pollution process have larger uncertainty, which may obscure the effect of controlled action.

In addition, in order to evaluate the environmental effectiveness of the control measures for atmospheric pollution, the spatial variations of pollutants concentrations and the source apportionment of PM2.5 were also analyzed in this study. The final conclusions were the results of the comprehensive consideration.

2. The error bars in Fig. 3 are too long, and thus the average values are uncertain. Some of the error bars are even longer than the reduction of concentrations caused by emissions controls (calculated by Eq. 9). For example, the error bar of PM2.5 concentrations during the period of CAHP (blue bar) in the whole city is much larger than the reductions compared to the NCAHP period (pink bar). There are many such cases in Fig. 3, as well as in Fig. 7, which makes the quantification analysis based on these data look not convincing.

Response: The error bars were the standard deviations of data in this study, which represented the fluctuation of data. In this paper, the monitoring sites of air quality distributed throughout the city of Shijiazhuang, which caused to the concentrations of air pollutants have large spatial differences (Fig. 1). In addition, Shijiazhuang city experienced several heavy pollution processes during the temporary emission control action (Fig. 2), likely leading to the concentrations of air pollutants have larger temporal differences. Therefore, the error bars are large in this study (as shown in Fig. 3). For the same reasons, the error bars of the chemical components in PM2.5 are also larger (Fig. 7).

As well known that the monitoring sites belonged to national, provincial and city controlling points (as seen in section 2.2.1 in paper), and all of them are managed in a standardized way. They all have QA/QC system. In addition, there is a strict quality control for the collection and analysis of the ambient PM2.5, and as shown in section 2.2 in paper. Therefore, the quality of the data itself is guaranteed in this study.

In this study, the decreased or increased values for the air pollutants or chemical species concentrations during different control stages were calculated at each monitoring sites, and the final results were obtained from the average values of these sites, so that the uncertainty of results can be further reduced in paper. The related contents have been added to supplemental material.

3. Some statement in the main text are very misleading. For example, in lines 366–367, "Well known that the NO2 is mainly derived from the vehicle exhaust (Liu et al., 2017b). Therefore, the control effect of motor vehicles was remarkable during the CAHP in Shijiazhuang." And in lines 391–392, the authors say "Furthermore, the effects of control measures for domestic coal might be worse during the CAHP." I don't think the data and analysis this paper presents can lead to such conclusions. The authors should be more precise what they find from their study.

Response: We agree with the views of the reviewer. The related statements have been removed in the revised manuscript. In this paper, some statements and related conclusions are not a causality to a large extent, but they do have some relevance. What is the cause of the relevance between them, and we will do a detailed study later.

In addition, the writing of English has also been modified to some extent.

Please also note the supplement to this comment:
https://www.atmos-chem-phys-discuss.net/acp-2017-1001/acp-2017-1001-AC2-supplement.zip
* * *
**Table 1.** The meteorological conditions during the four stages (NCANHP, NCAHP, CAHP and ACA) of the TECA period in Shijiazhuang.

|  | NCANHP | | NCAHP | | CAHP | | ACA | |
|---|---|---|---|---|---|---|---|---|
|  | Ave. | S.D. | Ave. | S.D. | Ave. | S.D. | Ave. | S.D. |
| Temperature (℃) | 8.4 | 3.6 | 7.4 | 2.4 | 3.1 | 3.8 | 0.7 | 2.7 |
| Relative humidity (%) | 77.7 | 17.0 | 73.4 | 15.7 | 71.5 | 18.0 | 83.3 | 18.1 |
| Wind speed (m/s) | 0.7 | 1.2 | 0.6 | 0.6 | 0.4 | 1.0 | 0.5 | 1.1 |
| Height of mixed layer (m) | 540 | 144 | 590 | 274 | 474 | 299 | 431 | 360 |

Ave. represents average value, S.D. represents standard deviation. NCANHP represents the no control action and no heating period, NCAHP represents the no control action and heating period, CAHP represents the control action and heating period, and ACA represents after control action.

**Fig. 1.**

$$P_{i-heating} = \frac{(C_{i-NCAHP} - C_{i-NCANHP}) \times 100}{C_{i-NCANHP}} \qquad (8)$$

$$P_{i-action} = \frac{(C_{i-NCAHP} - C_{i-CAHP}) \times 100}{C_{i-NCAHP}} \qquad (9)$$

where $P_{i\text{-}heating}$ represents the increasing percentage (%) of atmospheric pollutant concentration because of the combined effects of heating for cold winter and meteorological conditions; $P_{i\text{-}action}$ represents the decreasing percentage (%) of atmospheric pollutant concentration because of the combined influences of control action and meteorological conditions; $C_{i\text{-}NCANHP}$ represents the concentration ($\mu g/m^3$, CO: $mg/m^3$) of atmospheric pollutant during the no-control action and no-heating period; $C_{i\text{-}NCAHP}$ represents the concentration ($\mu g/m^3$, CO: $mg/m^3$) of atmospheric pollutant during the no-control action and heating period; $C_{i\text{-}CAHP}$ represents the concentration ($\mu g/m^3$, CO: $mg/m^3$) of atmospheric pollutant during the control action and heating period.

**Fig. 2.**

[Figure]

**Fig. 1.** Maps of the online monitoring stations and the filter membrane sampling sites in Shijiazhuang. The 24 online monitoring stations mainly include Twenty-second Middle School (TSMS), Fenglong Mountain (FLM), High-tech Zone (HTZ), Great Hall of the People (GHP), Century Park (CP), Water Source Area in the Northwest (WSAN), University Area in the Southwest (UAS), Staff Hospital (SH), Gaoyi (GY), Gaocheng (GC), Xingtang (XT), Jinzhou (JZ), Jingxing Mining District (JXMD), Lingshou (LS), Luquan (LQ), Luancheng (LC), Pingshan (PS), Shenze (SZ), Wuji (WJ), Xinle (XL), Yuanshi (YS), Zanhuang (ZH), Zhaoxian (ZX) and Zhengding (ZD). The filter membrane sampling sites are mainly located in TSMS, LQ and LC.

**Fig. 3.**

Fig. 2. The variations of atmospheric pollutants concentrations during the four stages (NCANHP, NCAHP, CAHP and ACA) of the TECA period in Shijiazhuang.

Fig. 4.

Interactive
comment

[Figure]

**Fig. 3.** The concentrations variations of PM$_{2.5}$, PM$_{10}$ and gaseous pollutants during the four stages (NCANHP, NCAHP, CAHP and ACA) of the TECA period in Shijiazhuang. Error bar represented standard deviation.

**Fig. 5.**

[Figure]

[Figure]

**Fig. 7.** The variations of chemical species in PM₂.₅ during the four stages (NCANHP, NCAHP, CAHP and ACA) of the TECA period. Error bar represented standard deviation.

**Fig. 6.**